# Affinity and Usage of Technology in General, and Telehealth in Particular amongst Michigan Older Adults during the COVID-19 Pandemic

**DOI:** 10.3390/healthcare12010117

**Published:** 2024-01-04

**Authors:** Zeenat Kotval-K, Isha Pithwa

**Affiliations:** School of Planning, Design & Construction, Michigan State University, East Lansing, MI 48824, USA; pithwais@msu.edu

**Keywords:** older adults, technology, telehealth, COVID-19

## Abstract

The COVID-19 pandemic and the travel restrictions imposed by many states led people to resort to technology for many of their daily needs which put older adults (aged 65 years and over) at a particular disadvantage as it is known that they are slow to adopt technology on a wide scale. Increasing the adoption and usage of technology for all purposes, especially healthcare appointments, would particularly benefit this population segment. Primary data was collected through online surveys targeted at older adults, aged 65 and over, living in Michigan, through a Qualtrics panel. Results indicate that, since this survey was an online survey, there is a bias in the use of technology as more than half the respondents had used zoom/skype or a similar medium to connect with friends and family during the period of travel restrictions. However, a substantial portion had not used telehealth services. Barriers to using telehealth services and factors that would encourage them to use them more are discussed. The study points to an emerging need for older adults to take advantage of technology more in order to overcome some of the barriers to accessing telehealth for their healthcare needs. Although technology cannot replace having real contact with people and being able to move about in the community, it helps, to a certain degree, to elevate overall wellbeing.

## 1. Introduction

Research on the aging population has suggested that although they increasingly prefer to age-in-place, they often suffer from physical, mental, and social isolation as they do not tend to drive around in their community as much as they once were able to when they were younger, due mostly to health concerns [1]. The COVID-19 pandemic has exacerbated this isolation as travel and mobility was not a personal choice but a mandated restriction during these times. Not only is technology an important factor in reducing isolation, it became particularly critical during the pandemic as healthcare appointments shifted to telehealth for many routine appointments. The advent of telehealth has transformed healthcare delivery, with particular relevance to older adults as it deals with having healthcare appointments online with a healthcare provider, without the need for an in-person office visit. As this demographic increasingly relies on telehealth services, it becomes imperative to scrutinize the unique challenges they encounter in adopting and utilizing this technology. The following literature review explores the intersectionality of social isolation, the need for telehealth, and the technological challenges faced by older adults.

### 1.1. The Use of and Access to Technology in Alleviating Social Isolation and Loneliness in Older Adults

The National Academies of Sciences, Engineering, and Medicine [2] conducted a comprehensive macro-study on social isolation and loneliness, emphasizing the impact on health and quality of life on adults aged 50 and older. The report recommends interventions that recognize the role of technology in addressing these issues, while acknowledging the disparities in technology adoption among older adults. Citing similar concerns, Kotwal et al. [3] stressed the interconnectedness between loneliness and the adoption of technology and that challenges exist for those who are not accustomed to or comfortable with digital platforms. Being accustomed to technology is also a function of availability, and research often underscores the challenges associated with the availability and adoption of technology, especially for older adults. Yu et al. [4] provide insights into the mediating effect of social contact on the relationship between internet use and loneliness. Their study suggests that internet use may be an effective tool for reducing loneliness by maintaining social contact. However, this raises concerns about the accessibility of the internet for older adults and their ability to utilize it for social interactions. Umoh et al. [5] explored the longitudinal association between technology use and social isolation among older adults. Their study indicates that access to technology is associated with a lower risk of social isolation. This finding suggests a potential solution that mitigates social isolation through technology, but also underlines the importance of equitable access for older adults.

Chen and Schulz [6] conducted a systematic review on the impact of information communication technology (ICT) interventions on reducing social isolation among the elderly. While the potential effectiveness of ICT is acknowledged, the review underscores that these interventions may not universally cater to the diverse needs of every senior. Khosravi et al. [7] further emphasized that while technology has the potential to alleviate social isolation, access remains a significant hurdle. Gardiner et al. [8] conduct an integrative review of interventions targeting social isolation and loneliness among older people. While various interventions show some success, the review emphasizes the need for adaptability and a community development approach. The role of technology in these interventions is underscored, pointing to its potential but also acknowledging the challenges in implementation.

### 1.2. Importance, Use, and Perceptions of Telehealth

The need for remote healthcare options was emphasized during the COVID-19 pandemic by researchers, such as Kotwal et al. [9], who have highlighted the prevalence of social isolation and loneliness among older adults and the increased need to manage these issues. Similarly, research by Howe et al. [10] investigated the relationship between social isolation, loneliness, and the utilization of telehealth among older adults. Socially isolated individuals tend to be less engaged with telehealth, raising concerns about accessibility to essential healthcare services.

Similar to arguments about access to and the availability of technology being an important point in the conversation on technology and loneliness, access to telehealth services is also an infrastructural element in healthcare that needs to be examined more closely. Bhatia et al. [11] and Baughman et al. [12] provide insights into the perspectives of older adults on telemedicine and compare the quality of in-person care with telehealth visits. A substantial proportion of older adults feel that telemedicine is inferior to in-person care, potentially influenced by technological challenges, thereby impacting healthcare access. Kalicki et al. [13] identified barriers to telehealth access among homebound older adults, revealing that confusion with the telehealth process is a significant obstacle. This technological barrier adds complexity to the challenges faced by older adults, particularly those with limited experience of or confidence in using digital platforms. Baughman et al. [12] assessed the quality of care between in-person and telemedicine visits. While their study supports the favorable association between the quality of primary care and telemedicine, it raises questions about the digital divide among older adults and how it may affect the overall quality of healthcare they receive.

### 1.3. Michigan Studies on Telehealth

Darrat et al. [14] explored socioeconomic disparities in patient use of telehealth, revealing that factors like age and income influence virtual visit completion. This suggests that older adults, particularly those in lower-income brackets, may face challenges in utilizing telehealth services. The Spartan Caregiver Support program [15] and family caregivers’ experiences with telehealth [16] provide a nuanced understanding of the challenges faced by caregivers and older adults in adopting telehealth. The concerns include issues related to access to technology, the relevance of telehealth for specific needs, and patient–clinician rapport in a virtual setting.

In summary, the importance of access to technology in general, and telehealth in particular, is imperative for maintaining health in older adults. This is important not only during a pandemic when they cannot travel, but also during normal times when they are advised to stop driving or driving becomes no longer feasible due to health reasons. This study therefore assesses the extent to which older adults in Michigan use technology for various reasons (email, communication, social interaction, shopping, etc.). The adoption of telehealth among older adults is a multifaceted process influenced by various concerns and technological challenges faced by this demographic. Recognizing and addressing these challenges is essential for ensuring equitable access to telehealth services, thereby contributing to improved healthcare outcomes and the overall wellbeing of older adults. To achieve this, it is important not only to assess the extent of the use of telehealth by older adults, but also to assess the perceptions of telehealth and challenges or barriers to using it. This aspect is under-represented in the research.

Consequently, this study aims to answer the following research questions: (1) To what extent do older adults in Michigan use technology? (2) To what extent do older adults use technology for their healthcare needs? (3) What are the barriers and enablers for using telehealth by older adults? The State of Michigan is used as the study area since it is the home state of the researchers and adds to the data on research with older adults that is already being conducted by the authors.

The results indicate the need for older adults to be able to adopt and use technology and devices for multiple purposes that would enable them to successfully age-in-place and remain physically, mentally, and socially active. In addition, the study investigates various socioeconomic characteristics and how they form factors affecting perceptions of telehealth.

## 2. Materials and Methods

This study uses a Qualtrics sample conducted with older adults aged 65 and over residing in the State of Michigan. The State of Michigan was chosen as the study area as it extends research already being conducted with older adults, relating to mobility and well-being, in the state by the authors. According to the 2020 census, Michigan has a total population of about 10 million, 17.2% (about 1.7 million) of whom are aged 65 years and over. A total of 55% of the older adults are female, 86% are White and 10% are Black or African American. Some more distinctive characteristics of older adults in Michigan include that 45% live alone, 33% have some form of disability, and 84% are not in the labor force. The Qualtrics panel method was chosen as the medium for acquiring responses as the response rates during the pandemic had been reportedly very low and the use of the panel would help the study achieve its sample size objectives. The study aimed to achieve a minimum sample size of about 385 as that would result in a 95% confidence level with an interval of 5% (survey system—sample size calculator). The Qualtrics online survey was conducted in late 2020 to early 2021 using a panel of respondents through the Qualtrics respondent database, where surveys are sent out to respondents based on the study’s needs. The study was approved by the University Institutional Review Board (IRB) as exempt (STUDY00004913). Informed consent was obtained from all respondents and all statistical analyses were conducted in IBM SPSS Statistics 27.

The survey itself asked respondents about demographic characteristics, living arrangements, use and comfort with technology in general, use of social media, online shopping habits, and use and perceptions of barriers to, and enablers for using telehealth. Using descriptive statistics and logistic regressions, the study aimed to assess the use of technology and perceptions of telehealth, in particular, among the older aged respondents.

## 3. Results

The survey received a total of 730 responses. Figure 1 shows the distribution of the survey respondents in Michigan, while Table 1 shows the demographic composition of the survey respondents. Figure 1 shows that there is a larger proportion of respondents from lower Michigan than the northern areas and a much lower representation of the upper peninsula region. One reason for this is that the lower half has a greater population density and greater number of urban areas than the north. Secondly, the southern areas have better internet availability, according to Connect Michigan, and therefore probably a higher number of respondents were able to participate in the online survey.

Table 1 shows an almost even split between gender and an average age of 70 for the respondents. The study has a slight overrepresentation of non-White respondents. In total, 80% of the respondents were White and 14% Black, while for the State of Michigan, 86% of the older adult population is White and 10% Black (U.S. Census, 2020). Almost a quarter of the respondents live alone while 62% live with someone else. Almost all respondents have insurance (health and prescription) and almost a quarter of the respondents have household incomes in the $20,000 to $40,000 and $40,000 to $60,000 categories.

This set of results aims to answer the first research question: To what extent do older adults use technology in general? Table 2 indicates that while three out of four respondents enjoy using technology, almost all of them have used it to check emails and to shop online, but only 18% have bought groceries online. Even with a substantial number of respondents enjoying using technology, only 60% have used it to connect with family and/or friends.

This set of results aims to answer the second research question: To what extent do older adults use technology for their healthcare needs? Table 3 shows that where using technology for healthcare needs is concerned, respondents have used it mostly to check lab (blood test) results, while only one in four have used it for telehealth purposes.

Diving more into the use of telehealth, these results aim to answer the third research question on what are the barriers to and enablers for using telehealth by older adults? Table 4 shows that respondents thought the biggest advantage of using it was that it saves time, which is understandable as their telehealth appointments are almost instant while obtaining an appointment to see one’s primary care physician often takes weeks. Respondents also thought the biggest disadvantage of using telehealth is that they cannot trust a diagnosis over the phone or online as they are not with the doctor face to face and the doctor cannot use other means to assess the patients’ health status. Respondents also worried about privacy and the security of their data where use of telehealth was concerned.

When those who did not use telehealth were asked what would encourage them to use it, a substantial 41% of the respondents said “Nothing” would encourage them to use it, while the largest percentage of respondents (75%) said they would if they were assured of privacy (see Table 5). In order to dig deeper into the characteristics of respondents that led to them thinking of certain advantages and disadvantages of telehealth, and what would encourage them to use it further, binary logistic regressions were performed, Table 6 shows the results of the regressions.

Since there were many binary logistic regressions carried out between various demographic factors and perceptions of telehealth, Table 6 shows only those regression results that were significant. Those with some college education were more likely to think that “Nothing” would encourage them to use telehealth than those with lesser educational attainment, i.e., no high school degree. Those who enjoy technology were less likely to think “Nothing” would encourage them to use telehealth compared to those that do not enjoy technology. That means that those that do not enjoy technology were more likely to think that nothing would encourage them to use telehealth, compared to those that do enjoy it. Those who enjoy technology were more likely to think ensuring greater privacy of information would be important in terms of encouraging them to use telehealth, compared to those that do not enjoy technology. Those with a high school degree were less likely to think training was an important factor that would encourage them to use telehealth than those without a high school degree. That means that those without the basic educational attainment of a high school degree would need to rely on training to be able to use technology to the level of telehealth use.

Males were less likely to think that having a webcam or good internet access were important factors in terms of encouraging them to use telehealth, compared to females. Older adults living in urban areas were less likely to think that having good internet access was an important factor in encouraging them to use telehealth, compared to those living in rural areas. Stated alternatively, those living in rural areas were more likely to worry about having stable internet connections for the use of telehealth than those living in urban areas. Those who lived alone were more likely to think “nothing” would encourage them to use telehealth than those who lived with someone else. Those with household incomes less than $20,000 were less likely to think “nothing” would encourage them to use telehealth, compared to those whose household income was more than $20,000. Essentially lower income adults were more amenable to using telehealth than upper income older adults. Those who lived alone were more likely to think telehealth saving them money was not an important factor in them not using it, compared to those living with someone else. So essentially, they were not worried about it saving them money. Those with a household income less than $20,000 were more likely to think that saving them money was an important factor contributing to them using telehealth, compared to those with a household income of greater than $20,000. Those who lived alone were less likely to think the disadvantage of no privacy was an important factor (since they lived alone), compared to those that lived with someone else.

## 4. Discussion

The results show a discrepancy between older adults using technology for various purposes. Given that over 40% of participants have not used technology to connect with friends and/or family, a concern over social isolation arises as has been shown by previous research [3,5]. And when 75% have not used telehealth to maintain healthcare appointments, a concern over health issues arises, similar to research carried out by Howe et al. [10]. It is understandable that issues with using telehealth point towards privacy and security concerns as healthcare is a sensitive topic and people want their privacy ensured.

On further investigation into what respondents thought were the pros and cons of using telehealth and what would encourage those that have not used it yet, some interesting dynamics have emerged. First, concerning what respondents thought were the pros or advantages of using telehealth, those with lower incomes were more likely to think telehealth saved them money, as was the case with those who lived alone. These results signify that the income-constrained and those living alone did worry about costs and that accessing doctors’ appointments from home was a more economic choice. On the other hand, those who lived alone also thought that issues of privacy were less of a concern to them than those who lived with another person. This is understandable as issues of privacy at their end would be a strong factor as they lived alone.

From the category of those that had not used telehealth services before, those that lived alone were more likely to think that there was nothing that would encourage them to use telehealth, a perception that those with higher incomes held as well. This points to the fact that, for those living alone and with higher incomes, there needs to be a greater and more concerted effort to motivate them to use telehealth as they are more adamant about not using it. Infrastructural capabilities, such as having better internet services and a webcam, were important considerations for those living in rural areas and for females. This is a more nuanced finding as to what specific elements of the older adult population are concerned with when it comes to access to infrastructure than the general findings by Raj et al. [16]. This perception from rural residents is understandable as internet availability is better in urban areas. However, the perception of such capabilities by females is less understood. One possibility could be that males were more likely to have access to webcams and the internet already compared to females, a clear structural inequity component.

Where education is concerned, those without a highs school degree were more likely to think that formalized training on the use telehealth would be an important factor in them using it, while they were less likely to think that nothing would encourage them to using telehealth. This indicates that they would be more amenable to using telehealth if they had proper training to do so. Similarly, those who already enjoy using technology were less likely to think that nothing would encourage them to use telehealth as they were already amenable to using technology and enjoyed doing so, a finding relatable to that from Kalicki et al. [13]. They also thought more about privacy concerns, and this could point to the fact that, since they enjoy using technology, they are more aware of its disadvantages and ensuring privacy was a significant issue that would encourage them to use telehealth.

This study has some limitations that should be acknowledged. First, there is an inherent bias towards those with access to technology in some form as this was an online survey. There should also have been oversampling in the northern regions of Michigan as the sample from this region is much less represented, although this region has lower densities than the southern half of the state. A follow-up study towards the end of the pandemic might have helped us learn whether older adults adopted telehealth to a greater degree during the COVID-19 pandemic, especially since the survey was conducted in the first year of the pandemic.

## 5. Conclusions

While this study has corroborated some of the findings from research previously carried out on the use of technology and telehealth by older adults, it lends some nuanced perspectives to move this research direction further. This study presents the perceptions of older adults in Michigan on why they use (or do not use) telehealth, and what aspects of this healthcare avenue they feel are beneficial or detrimental to them. This study also presents their opinions on what would encourage them to use telehealth to a greater degree. This is an important part for healthcare delivery as knowing these aspects of telehealth delivery could help providers convince more of their older adult clients to use this form of healthcare and help tremendously in maintaining overall health. The implications of this research lead to suggesting enhanced efforts to acquaint older adults more with technology so they can be socially connected to friends and family such as, for example, the GetSetUp Program in Michigan [17]. Noticing the lasting health effects of isolation and loneliness, the Michigan Department of Health & Human Services has teamed up with the GetSetUp Program to offer online classes to the older adult community to help them remain physically, mentally, and socially active. Healthcare workers, caregivers, and case managers can help older adults with a few telehealth appointments to become comfortable. This would address the response from the participants who said they would be more likely to use telehealth if they were shown how to do it or if they had some training for it. Insurance companies could help older adults obtain specific devices that would be used for doctors’ appointments only. Insurance companies such as Medicaid in Michigan have started to become involved with the social determinants of health by providing, for example, transportation/taxi rides/reimbursement for transportation costs for clients traveling to their regular doctor’s visits as it is known that this is more effective in terms of maintaining health and, therefore, less of a monetary drain for those companies [18]. While no amount of technology can solve the issue of physical and social isolation, technology can be helpful when travel is restricted for any reason, be it a pandemic or through driving cessation. Missing doctors’ appointments creates compounding health issues that can be avoided through maintaining regular preventive care appointments.

## Figures and Tables

**Figure 1 healthcare-12-00117-f001:**
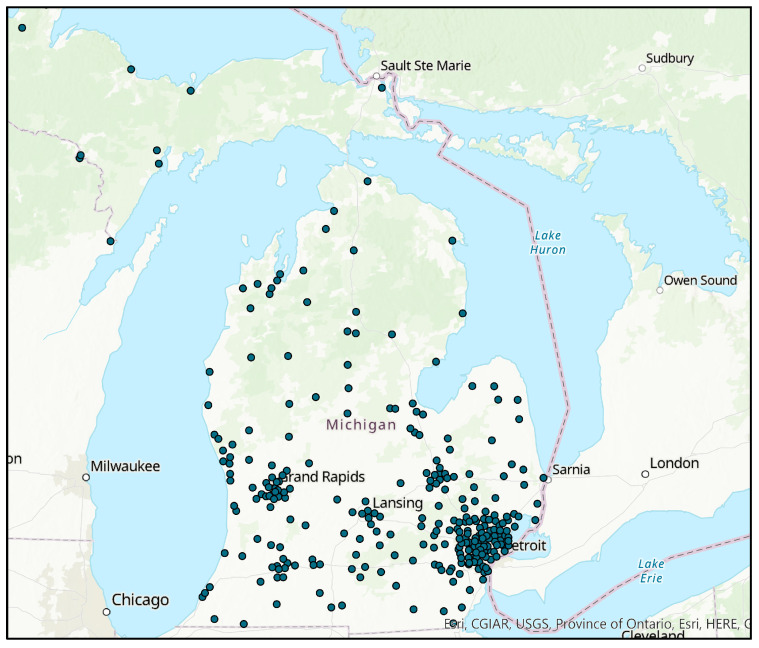
Location of Study Respondents in Michigan.

**Table 1 healthcare-12-00117-t001:** Demographic results of study participants.

Description	Percent of Respondents
Average Age	70
Male	53%
White (Black)	80% (14%)
Retired (Working)	77% (11%)
Household Size of 2 (1)	62% (26%)
Live in Own Home (Condo)	78% (10%)
Married	58%
Live in Urban Areas	68%
Still Drive Themselves	93%
Have Medical (Prescription) Insurance	98% (91%)
Annual Household Income Under $20,000	11%
Annual Household Income $20,000–<$40,000	26%
Annual Household Income $40,000–<$60,000	23%
Annual Household Income $60,000–<$80,000	14%
Annual Household Income $80,000–<$100,000	9%
Annual Household Income ≥ $100,000	17%

N = 730.

**Table 2 healthcare-12-00117-t002:** Use of Technology for Personal Use.

Description	Percent of Respondents
Enjoy using technology	74%
Use technology for emails (Facebook)	97% (47%)
Used zoom/skype	59%
Shopped online	93%
Bought groceries online	18%

**Table 3 healthcare-12-00117-t003:** Use of Technology for Healthcare Needs.

Description	Percent of Respondents
Ordered prescriptions online	43%
Booked an appointment with provider	28%
Sent a message to the doctor	41%
Checked lab results	62%
Used Telehealth	25%

**Table 4 healthcare-12-00117-t004:** Perceptions of Telehealth.

Description	Percent of Respondents
Pros/Advantages	
Saves money	34%
Saves time	83%
No wait times	75%
Cons/Disadvantages	
No privacy	48%
Worry about data security	61%
Don’t trust a diagnosis over the phone	66%

**Table 5 healthcare-12-00117-t005:** What might encourage respondents to use telehealth.

Description	Percent of Respondents
Nothing!	41%
If someone sat with me and explained it	49%
If I’m assured of privacy	75%
If I got trained with a similar group	34%
If I had a webcam	39%
If I had reliable and better internet service	30%

**Table 6 healthcare-12-00117-t006:** Binary Logistic Regressions of Perceptions of Telehealth Use.

Independent Variable	B (Coeff)	S.E.	Exp(B)
Encourage: Nothing (Reference No High School Degree) Negelkerke R^2^ = 0.049, N = 530	0.598 **	0.196	1.818
Encourage: Nothing (Ref: Do not enjoy technology) Negelkerke R^2^ = 0.049, N = 530	−1.177 *	0.518	0.308
Encourage: Privacy (Ref: Do not enjoy technology) Negelkerke R^2^ = 0.014, N = 489	1.194 *	0.578	3.301
Encourage: Training (Ref: No High School Degree) Negelkerke R^2^ = 0.018, N = 389	−0.455 *	0.229	0.635
Encourage: Webcam (Ref: Females) Negelkerke R^2^ = 0.011, N = 448	−0.369 *	0.190	0.692
Encourage: Good Internet (Ref: Females) Negelkerke R^2^ = 0.013, N = 462	−0.407 *	0.195	0.666
Encourage: Good Internet (Ref: Rural) Negelkerke R^2^ = 0.025, N = 462	−0.594 **	0.202	0.552
Encourage: Nothing! (Ref: Lived with another) Negelkerke R^2^ = 0.009, N = 528	0.371 *	0.196	1.450
Encourage: Nothing! (Ref: Household Income >$20,000) Negelkerke R^2^ = 0.059, N = 529	−0.638 *	0.275	0.528
Pro: Saved Money (Ref: Lived with another) Negelkerke R^2^ = 0.054, N = 179	−1.068 **	0.429	0.344
Pro: Saved Money (Ref: Household Income >$20,000) Negelkerke R^2^ = 0.014, N = 175	2.153 *	1.046	8.614
Con: No Privacy (Ref: Lived with another) Negelkerke R^2^ = 0.034, N = 179	−0.763 *	0.362	0.466

* *p* < 0.05; ** *p* < 0.01; *** *p* < 0.001.

## Data Availability

The data presented in this study are available on request from the corresponding author. The data are not publicly available due to privacy reasons.

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
