# Peer review of "Affinity and Usage of Technology in General, and Telehealth in Particular amongst Michigan Older Adults during the COVID-19 Pandemic"

_healthcare, 2024, doi:10.3390/healthcare12010117_

Round 1

Reviewer 1 Report

Comments and Suggestions for Authors

I think this paper is very interesting.

At a time when the importance of telemedicine is emerging due to COVID-19, we believe that this is an essential study in that it is an observational study to increase the elderly's awareness and adaptability to telemedicine.

Let me ask you a question.

1. Please explain in more detail the method or type of telemedicine you wish to mention in this study.

And please insert the explained part into the text.

2. The content of the discussion lists only the author's thoughts. The discussion should highlight the originality of this paper compared to previously published research papers. So please explain why you didn't do that and let us know how you plan to do so.

Author Response

Dear Reviewer,

Detailed Responses to Reviewer Comments:

Reviewer 1

  1. Please explain in more detail the method or type of telemedicine you wish to mention in this study. And please insert the explained part into the text.

Response: Thank you for pointing this out.  We have clarified what telehealth we refer to in the first paragraph of the Introduction section.

  1. The content of the discussion lists only the author's thoughts. The discussion should highlight the originality of this paper compared to previously published research papers. So please explain why you didn't do that and let us know how you plan to do so.

Response: Thank you for this comment.  The discussion section has been revised linking the findings to previously conducted research. 

Reviewer 2 Report

Comments and Suggestions for Authors

Thank you for your manuscript. Overall, it seems like an interesting study but the reasons for undertaking this work are not clear. The title does not reflect the aims or the findings of the study. Can you generalise based on your findings only? Are you focusing on telehealth or technology use? It is unclear what your paper is about from the title.

The study lacks clarity around the aims. Why is this study important? What value will it add? The introduction does not support the aims well. You need to set the scene with the introduction and then use the aims to justify why you are doing the study. Currently, it does not flow well.  

Older adults/ageing population is not well defined

Line 27 - socially, should be social

Line 26-29 - Where is the evidence to suggest that older people do not move around as much compared to when they were younger? This either needs a reference, or better context. 

Line 39 - 69 - This paragraph seems more about accessibility than use. Perhaps rephrase the subtitle.

Line 71 - 73 - This sentence fits more closely with the first paragraph (above). 

Lines 94 -112 - This paragraph is not about Michigan. If it is, it needs to be referenced back to this community. The subtitle seems incorrect. You have not justified why it is important to undertake this study in Michigan. Where is the context? Where is the information about the local community and its needs? What are the demographics? 

Line 118 - 122 - Results do not belong in the introduction section. 

Results: The tables are clear but the written results need revision.  Please do not use dot points and do not discuss the results in this section (this should be done in the discussion). Lines 199 - 240 could probably all be in the discussion, with just some major overarching information in the results. 

Discussion: The discussion has no reference back to supporting evidence. How does social isolation link to not using technology to contact family and friends? Perhaps they live close-by? You need to justify your standing on your findings. You need to link your discussion back to evidence and other literature. 

Conclusion: Your conclusions do not align well with your aims. And you have made unsupported statements such as insurance being able to help. You have no evidence of this. You conclusion needs to be based on your aims and your findings. 

Comments on the Quality of English Language

The language will need to be reviewed and revised by an English speaking person. There is a lot of terminology that is incorrectly used and repeated. 

Author Response

Dear Reviewer,

Detailed Responses to Reviewer Comments:

Reviewer 2

  1. Thank you for your manuscript. Overall, it seems like an interesting study but the reasons for undertaking this work are not clear. The title does not reflect the aims or the findings of the study. Can you generalize based on your findings only? Are you focusing on telehealth or technology use? It is unclear what your paper is about from the title.

Response: Thank you for your comment.  In fact, the title says exactly what this study aimed to find out.  We have mentioned the use of technology in general and telehealth in particular so we are focusing on both aspects and have shown results on both aspects as well. 

  1. The study lacks clarity around the aims. Why is this study important? What value will it add? The introduction does not support the aims well. You need to set the scene with the introduction and then use the aims to justify why you are doing the study. Currently, it does not flow well.  

Response: We have revised the paragraph preceding the aims paragraph on Page 3 to emphasize the importance of the study and the research questions it aims to address.  We hope this reads better now.

  1. Older adults/ageing population is not well defined

Response: Thank you for catching that.  We have defined that first in the Abstract and then again in the Methodology section.

  1. Line 27 - socially, should be social

Response: Thank you for catching that.  This correction has been made.

  1. Line 26-29 - Where is the evidence to suggest that older people do not move around as much compared to when they were younger? This either needs a reference, or better context. 

Response: Thank you for pointing this out.  We have revised it to say they do not tend to drive around as much when they were younger due mostly to health concerns.  We have cited this as well.

  1. Line 39 - 69 - This paragraph seems more about accessibility than use. Perhaps rephrase the subtitle.

Response: Thank you for pointing this out.  We have revise the subtitle to include “Access” but have kept the term “Use of” because this section deals with both facets of technology.

  1. Line 71 - 73 - This sentence fits more closely with the first paragraph (above). 

Response: This study is referenced in the above section, but we include it here as well to show the connection between loneliness and isolation and telehealth use to maintain mental health.

  1. Lines 94 -112 - This paragraph is not about Michigan. If it is, it needs to be referenced back to this community. The subtitle seems incorrect. You have not justified why it is important to undertake this study in Michigan. Where is the context? Where is the information about the local community and its needs? What are the demographics? 

Response: Respectfully, the studies mentioned in this paragraph are all about Michigan. This paragraph is included to show the research that has been done in Michigan specifically as it is the State where this study concentrated.  We have, however, added more context in the Methodology section where we put in more context to the Michigan Study Area.

  1. Line 118 - 122 - Results do not belong in the introduction section. 

Response: We followed the directions for the journal from the homepage and the Guide for Authors page and that is why we included the main aspects of the results here.

  1. Results: The tables are clear but the written results need revision.  Please do not use dot points and do not discuss the results in this section (this should be done in the discussion). Lines 199 - 240 could probably all be in the discussion, with just some major overarching information in the results. 

Response: We have removed the bulleted points and left the explanation of how to read the results in Table 6 as paragraphs.  The discussion section already has the discussion on these points so we have not moved the bullet points to that section as they are already discussed there.  We have also tried not to discuss the results in this section. 

  1. Discussion: The discussion has no reference back to supporting evidence. How does social isolation link to not using technology to contact family and friends? Perhaps they live close-by? You need to justify your standing on your findings. You need to link your discussion back to evidence and other literature. 

Response: Thank you for this comment.  The discussion section has been revised linking the findings to previously conducted research. 

  1. Conclusion: Your conclusions do not align well with your aims. And you have made unsupported statements such as insurance being able to help. You have no evidence of this. You conclusion needs to be based on your aims and your findings. 

Response: Thank you for this comment.  We have also revised the conclusion section to provide context and evidence of the linkages we propose.

Round 2

Reviewer 1 Report

Comments and Suggestions for Authors

The author made his own revisions in accordance with the reviewer's opinions, but they were not sufficient. However, since there are limitations to the study as described, I will not ask any further questions. Thank you for your hard work.